# Predictive Value of MRI Pelvimetry in Vaginal Delivery and Its Practicability in Prolonged Labour—A Prospective Cohort Study

**DOI:** 10.3390/jcm12020442

**Published:** 2023-01-05

**Authors:** Juan Li, Ying Lou, Cheng Chen, Weizeng Zheng, Yuan Chen, Tian Dong, Mengmeng Yang, Baihui Zhao, Qiong Luo

**Affiliations:** 1Department of Obstetrics, Women’s Hospital, Zhejiang University School of Medicine, Hangzhou 310006, China; 2Key Labouratory of Women’s Reproductive Health of Zhejiang Province, Women’s Hospital, Zhejiang University School of Medicine, Hangzhou 310006, China; 3Department of Radiology, Women’s Hospital, Zhejiang University School of Medicine, Hangzhou 310006, China

**Keywords:** MRI pelvimetry, predictive model, vaginal delivery, prolonged labour, nomogram

## Abstract

Background: Pelvic dimensions are crucial variables in the labour process. We used magnetic resonance imaging (MRI) pelvimetry to predict the probability of vaginal delivery and distinguish the cephalopelvic disproportion risk in women with prolonged active labour. Methods: This prospective cohort study enrolled term nulliparous women willing to undergo MRI pelvimetry and a trial of labour. A nomogram, with vaginal birth as the outcome, was developed and evaluated by calibration curve and decision curve analyses. The pairwise association between maternal and fetal parameters and a prolonged first stage of labour was quantified. Results: Head circumference (HC), abdominal circumference (AC), intertuberous distance (ITD), interspinous diameter (ISD), and body mass index (BMI) were introduced to develop a nomogram with good diagnostic performance (area under the curve = 0.799, sensitivity = 83%, and specificity = 73%). The cephalopelvic index of diameter (CID) in 54 women with a prolonged first stage of labour was much smaller in those who delivered via cesarean section compared with those who delivered vaginally (18.09 ± 1.14 vs. 21.29 ± 1.06; *p* = 0.046). Conclusions: An MRI pelvimetry-based nomogram may predict the probability of vaginal delivery. Practitioners should reassess the pelvimetry parameters to decide whether the trial of labour should be continued if it is prolonged.

## 1. Introduction

A safe delivery that avoids adverse outcomes for both the mother and baby is important. A major factor that influences whether vaginal delivery is successful or not is fetal and pelvic compatibility [1]. The pelvic capacity enables the fetus to pass through the birth canal [2], thus, pelvimetry can help a clinician to decide whether to offer an elective cesarean section to a woman due to the risk of cephalopelvic disproportion (CPD).

Pelvimetry relies on imaging techniques, but X-ray and computed tomography pelvimetry are not ideal methods for pregnant women, as they expose the fetus to ionizing radiation and increase the risk of childhood cancer [3,4]. Magnetic resonance imaging (MRI) pelvimetry in pregnant women, first described by Stark et al. [5], may be a promising imaging modality. It is safe during pregnancy because of its minimal radioactivity and pelvimetric errors of approximately 1% [5]. Some researchers have indicated that hormonal changes during pregnancy may contribute to pelvic ligament laxity, which causes pelvic bone separation during delivery, thus increasing pelvic dimensions [6,7]. However, other researchers have proposed the different idea that pelvic dimensions do not change during pregnancy [8]. Currently, due to this concern about the change in pelvic dimensions, our hospital mainly performs MRI pelvimetry late in the third trimester to minimize errors in assessing pelvic adequacy. The financial cost makes the use of pelvic MRI in China limited, so there is a lack of well-established pelvic MRI data for Chinese people. Some pregnant women with good MRI pelvimetry-based parameters may still have adverse pregnancy outcomes and the reasons for this should be evaluated further.

Our study evaluates the multiple MRI pelvimetry-based factors influencing successful vaginal deliveries and seeks to establish an appropriate decision-making strategy for deliveries. We focused on women with a prolonged first stage of labour who ultimately delivered vaginally or via cesarean sections. Pelvic dimensions can be a practical and feasible way of helping practitioners change the delivery mode in time to reduce the risk of adverse outcomes.

## 2. Materials and Methods

This prospective cohort study was conducted between 2020 and 2021 in China. Approval was obtained from the ethics committee at the Women’s Hospital, School of Medicine Zhejiang University (IRB-20200044-R). We calculated the sample size required for the prediction model using an established rule of thumb that ensures at least 10 events for each predictor parameter [9]. Pregnant women willing to evaluate the matching degree of pelvis and fetus were enrolled. We eventually enrolled 215 nulliparous women with singleton term gestations. They underwent MRI pelvimetry between 37 to 39 weeks of gestation. The women had fetuses in either the cephalic or breech presentation. Women with fetuses in breech presentations intended to have a vaginal delivery after an external cephalic version. None of the women had contraindications for MRI imaging or fetuses with lethal congenital malformations. They all gave informed consent at enrollment. All pregnant women underwent a trial of labour, which means women without any severe complications for vaginal delivery who should try to deliver vaginally no matter if spontaneous or by induction. The exclusion criteria were as follows: (1) Women who were unable to deliver vaginally due to adverse complications, such as placental abruption and pre-eclampsia; (2) Women who underwent cesarean section without a trial of labour; (3) Women with suspected intrapartum fetal compromise on fetal electrical monitor or ultrasound; (4) Women who did not give birth at our hospital (Figure 1).

MRI pelvimetry measurements included the obstetric conjugate, intertuberous distance (ITD), interspinous diameter (ISD), pubic angle, pelvic width, bilateral acetabulum distance, sacral outlet diameter, coccygeal pelvic outlet, sacral length, and pelvic inclination (Figure 2). The fetal size was determined by the last prenatal ultrasound done within one week before expected date of delivery. The data on maternal and fetal outcomes and duration of labour were also collected.

During the examination of the bony pelvis (1.5-T MRI scanner, Signa HDxt; GE Healthcare, Milwaukee, WI, USA), the women were in a supine position for 10 to 15 min. All digital images were analyzed using a software picture archiving and communication system (version 1.0.0.25909) (Greenlander, Zhejiang Greenlander I.T. Co., Ltd., Hangzhou, Zhejiang, China). Two radiologists with at least five years of experience in gynecological MRI performed the investigation independently and the measurements were assessed for inter-observer reliability.

The cephalopelvic index of diameter (CID) is defined as the difference between the mean diameter of the midpelvis and the fetal BPD. It is calculated by the formula (OC + ISD)/2-BPD, which represents the relative fit of the fetal head and the maternal pelvis [10]. Women with CPD were defined as those with normal uterine contractions (at least two or more spontaneous contractions per 10 min within the last hour or longer) who experienced cervical arrest over 4 h after the cervix had reached 4 cm, and those who had protracted descent of at least 2 h after the cervix had dilated to 10 cm. All CPD patients underwent cesarean section. A prolonged first stage of labour based on patterns in the Chinese population was defined as a cervical arrest of over 4 h after reaching 3 cm in the presence of adequate contractions [11]. The practitioners managing labour were blind to the MRI results to avoid bias when choosing the delivery mode. 

Continuous variables were presented as the mean ± standard deviation, and categorical variables were summarized as counts and percentages. The Shapiro–Wilk test was used to measure the normality of the distribution of continuous variables. Differences between pelvimetric dimensions were tested for significance using Student’s *t*-test and the Mann–Whitney U test. The Chi-squared test was used to evaluate nominal variables. Collinearity analysis was performed on all independent variables to eliminate those with a variance inflation factor of greater than 10.

Intraclass correlations (ICCs) were introduced to verify the inter-and intra-examiner variability [12]. ICCs were calculated for each MRI parameter. We interpreted an ICC greater than or equal to 0.75 as having good consistency.

The study population was divided into the training set (*n* = 75) and validation set (*n* = 75) using blocked randomization to ensure there was an equal number of prolonged labour cases in both groups. Univariable logistic regression analysis was performed to estimate the associations between the predictive variables and delivery outcomes, which were the fundamentals of the model after the adjusted odds ratios (ORs) and 95% confidence intervals (CIs) were calculated, then the nomogram was constructed using multivariable logistic regression. Each predictor point was first determined by drawing a vertical line to the points axis and then the points of each predictor were added. Finally, a line was drawn from the total point axis to determine the estimated vaginal delivery probabilities at the lower line of the nomogram. The score was applied to all women in the validation set, and a receiver operating characteristic (ROC) curve analysis was used to evaluate the vaginal birth prediction. We did a calibration curve to detect the concentricity between the model probability curve and the ideal curve. A decision curve analysis was also applied to evaluate the net benefit of the model. All tests were two-sided and had an alpha level of 0.05.

Women with a prolonged first stage of labour were divided into those who delivered via cesarean sections and those who delivered vaginally. We calculated the CID cutoff value of vaginal delivery in our study and then compared the demographic and MRI pelvimetry data between the cesarean section and vaginal delivery groups. The statistical analyses were performed using SPSS 26.0 (IBM Corp., Armonk, NY, USA) and R (http://www.r-project.org, accessed on 26 April 2022). A *p* value of less than 0.05 was considered statistically significant.

## 3. Results

The maternal age of the cohort was 30.3 ± 3.7 years old, gestational age was 39.3 ± 0.7 weeks, and the newborn birth weight was 3454.8 ± 396.5 g. The pre-pregnancy body mass index (BMI) was 26.5 ± 3.0 kg/m^2^, and 19 (13%) women had obesity. A total of 62 (41%) pregnant women delivered via cesarean sections, whereas 88 (59%) delivered vaginally. The 150 eligible pregnant women were split into the training (*n* = 75) and validation sets (*n* = 75). The maternal demographic data, labour information, neonatal outcome, and MRI parameters for the two groups are presented in Table 1. None of the maternal characteristics and neonate information differ between the two sets. All MRI dimensions with inter-and intra-observer ICCs greater than or equal to 0.8 indicated favorable stability for radiomics feature extraction.

Univariable logistic regression analyses indicated factors associated with vaginal birth (Table 2), and multivariate logistic regression introduced five factors, including HC, AC, ITD, ISD, and pre-pregnancy BMI (Table 3), which we used to develop a nomogram that predicts the vaginal delivery probability (Figure 3). 

Women with lower pre-pregnancy BMI, bigger ISD and ITD, and fetuses with smaller HC and AC increased the likelihood of vaginal birth. Collinearity analysis was performed to exclude the dependent relationships. The ROC curve analysis measured the performance of this nomogram, and the AUC of this model was 0.799 in the validation set (Figure 4), indicating good diagnostic performance with a sensitivity of 83% and a specificity of 73% at the optimal cutoff value. A calibration curve demonstrated that our model showed good fit and calibration with the ideal curve (Figure 5). Our decision curve analysis demonstrated a favorable clinical effect of the predictive model (Figure 6).

There were 54 pregnant women with a prolonged first stage of labour based on the standard mentioned in the Methods section. We separated them into the cesarean section group (*n* = 25) and vaginal birth group (*n* = 29) according to the experienced practitioners’ decision, mostly due to CPD risk. We found that fetal BPD and AC were significantly increased for the women who eventually had a cesarean section (*p* < 0.05). The CID reflects the fit of the fetal head and the maternal pelvis. The cutoff value of CID calculated was 19.8, and women with a CID less than or equal to 19.8 were more likely to deliver by cesarean section, with a sensitivity of 78% and a specificity of 67%. The CID value of the cesarean section group was much smaller than the CID value of the vaginal birth group (18.09 ± 1.14 vs. 21.29 ± 1.06; *p* = 0.046). Pregnant women with a smaller pubic angle were at a greater risk of a cesarean section, but there was no statistically significant difference (Table 4). 

## 4. Discussion

We constructed an MRI pelvimetry-based prediction model for the probability of vaginal delivery. We found that women with lower BMIs, bigger ISD and ITD dimensions, and fetuses with smaller HC and AC dimensions had a higher probability of vaginal delivery. We found that BMI, HC, and ISD might, therefore, be the most relevant to vaginal delivery based on the nomogram. Our model was further validated with good performance. The MRI pelvimetry-based CID cutoff value of vaginal delivery in our research was 19.8. To find a favorable pelvimetry parameter to distinguish the likelihood of CPD, we focused on women with a prolonged first stage of labour. Our results demonstrated that women with greater CID values, smaller fetal BPD, and AC should be allowed to continue trying labour under close monitoring. Also, our results revealed that MRI pelvimetry could be a valuable tool for selecting nulliparous women who can undergo a trial of labour. We are working hard to apply MRI pelvimetry to more pregnant women who can undergo a trial of labour. Because of the high costs involved, at the very least, it should be considered in pregnant women at risk of CPD.

Cesarean delivery rates approach 30% to 40% in China [13], which comes with associated morbidity and greater use of resources. Researchers have committed their time to establish a validated tool that can predict a successful vaginal birth. Factors related to a safe vaginal delivery include an adequately sized and shaped maternal bony pelvis, soft-tissue shaping of the birth canal during delivery, sufficient uterine contractions, and a fetal head of the proper size with the ability to mold [14]. The progress of labour in Chinese women is slower than that of their Western counterparts [11]. Therefore, we need an effective model suitable for clinical practice in the Chinese population.

The fetus–pelvis fit remains a priority during vaginal delivery. Previous studies have demonstrated that MRI can be reliable in evaluating labour outcomes [15,16,17]. Some researchers have attempted to validate MRI pelvimetry and its prediction of the success of vaginal delivery in women with previous cesarean sections but were limited by the small cohort size [18]. Of all the MRI pelvimetry parameters, an obstetric conjugate is the most valuable predictive pelvic metric because of its strong positive correlation with successful vaginal delivery [19]. In one study, scholars used an obstetric conjugate of 12 cm to represent a sufficient pelvic inlet for the trial of labour [20]. Another research proposed a minimum obstetric conjugate of 10 cm for a fetus weighing 3400 g to pass through [21]. However, various racial differences limited the applicability of the data.

A higher BMI may influence soft-tissue pelvic volume that is significantly associated with dystocia [19]. Our results indicated that women with a lower BMI had an increased vaginal birth probability. Additionally, maternal pelvimetry ISD and ITD, and fetus HC and AC are strongly associated with the success of vaginal deliveries. The ISD is the parameter that represents the midpelvis, and its stenosis usually results in obstructed internal rotation of the fetus that affects the head’s descent [22]. Our findings indicate that a smaller ISD is a risk for cesarean section, and the pelvic outlet ITD also plays an important role in the probability of vaginal delivery.

We introduced the CID that was previously used in ultrasound and innovatively proposed in MRI. Our results clarified that women with term pregnancies may benefit from planned cesarean sections when their CID was below the threshold value of 19.8, but further prospective validation is needed. During prolonged labour, decision-making strategies are important. Our results show that women in the cesarean section group carried larger fetuses and their CID was significantly smaller than that of those in the vaginal delivery group. Therefore, MRI pelvimetry can be a reliable adjunct tool in evaluating the risk of CPD in the prolonged first stage of labour. Based on this evaluation, women would benefit from early referral for cesarean section.

### Strengths and Limitations

Our results demonstrated the high inter-and intra-examiner reliability of MRI pelvic measurements. Our radiologists were very experienced and hence the precision and reliability of the MRI imaging.

However, there are some shortcomings in our research. First, pregnant women were assessed by MRI in the supine position only. Previous studies indicate that MRI pelvimetry showed an increase in pelvic parameters when women were in the upright position [23,24]. Gravity also causes the descent of the fetal head, with pressure directly and evenly on the cervix, leading to more regular and frequent contractions [16]. Hence, women with borderline CPD who maintain an upright position may reduce the risk of an emergency cesarean section. However, most women in our hospital were in a low semi-recumbent position during labour, which may limit the results. We should also notice that the grand majority of shoulder dystocia incidents occur in patients with normal pelvic dimensions; MRI pelvimetry would not be useful for identifying patients at increased risk as traditional pelvimetry. Additionally, this study was limited by the relatively small sample size due to the high costs of MRI pelvimetry, as well as the limited geographic area. For this reason, the external validity of our study needs further investigation through a well-designed, randomized-control trial.

Other factors, including the degree of pelvic inclination and the angle of fetal entry into the pelvis, cannot be judged by in-plane measurements alone. Our research group is cooperating with artificial intelligence management to reconstruct pelvic imaging in 3D diagrams that imitate real scenarios for future research.

## 5. Conclusions

In summary, our study indicated that an MRI pelvimetry-based model could be a reliable method applied to predict vaginal delivery in clinical practice. When women have a prolonged first stage of labour, practitioners should reassess the pelvimetry parameters to decide whether it is worth continuing the trial of labour. Our MRI pelvimetry findings provide a better understanding of the pelvic characteristics in Chinese women that should be considered when making clinical decisions.

## Figures and Tables

**Figure 1 jcm-12-00442-f001:**
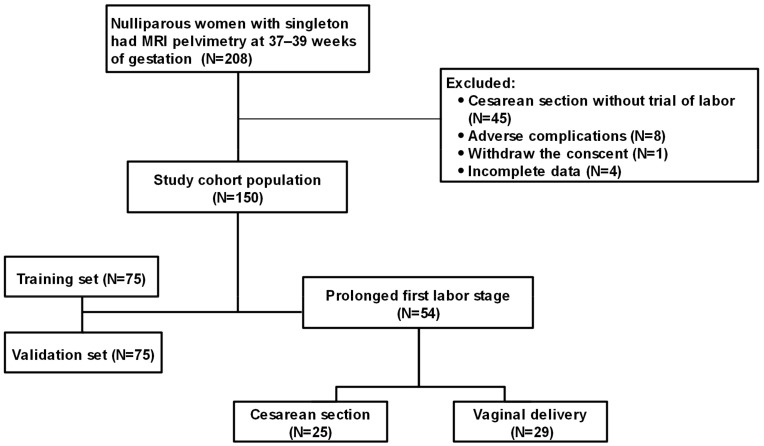
The study participant flow diagram. Nulliparous women with singleton gestation had MRI pelvimetry (excluding those with comorbidities who underwent cesarean section without trial of labour) at 37 to 39 weeks of gestation. A study cohort of 150 pregnant women was first divided into the training and validation sets, then those with a prolonged first stage of labour were classified into the cesarean section and vaginal delivery groups.

**Figure 2 jcm-12-00442-f002:**
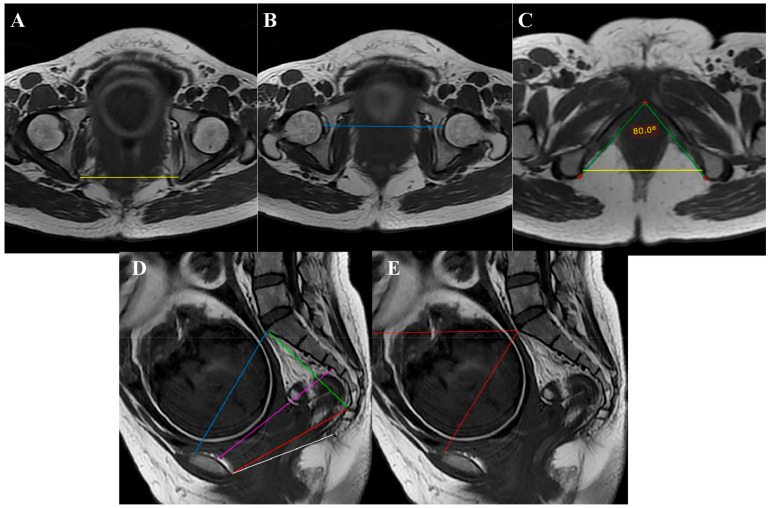
MRI pelvimetry: (**A**) Intertuberous distance; (**B**) Bilateral femoral head distance; (**C**) Pubic angle (green line) and interspinous distance (yellow line); (**D**) Obstetric conjugate (blue line), pelvic width (purple line), sacral outlet diameter (red line), outlet diameter of the pelvis (white line), and sacral length (green line); (**E**) Pelvic inclination (red line).

**Figure 3 jcm-12-00442-f003:**
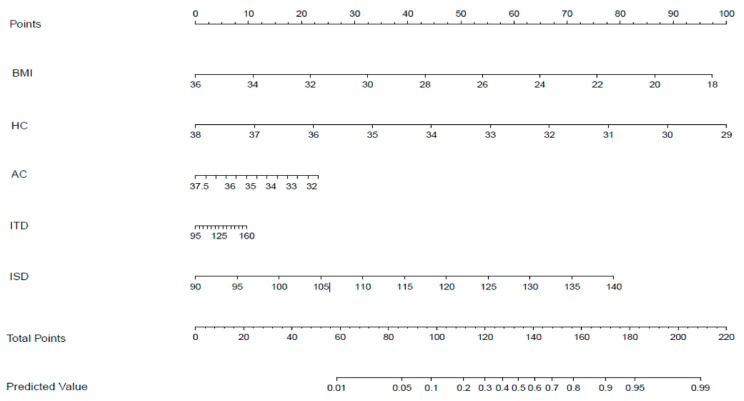
A predictive nomogram for the probability of vaginal delivery. The nomogram is used as follows: (1) to obtain the nomogram-predicted probability, locate values on each axis; (2) draw a vertical line to the point axis to determine how many points are attributed for each variable value; (3) sum the points for all variables; (4) locate the sum on the total point line to assess the vaginal delivery probability at the lower line of the nomogram.

**Figure 4 jcm-12-00442-f004:**
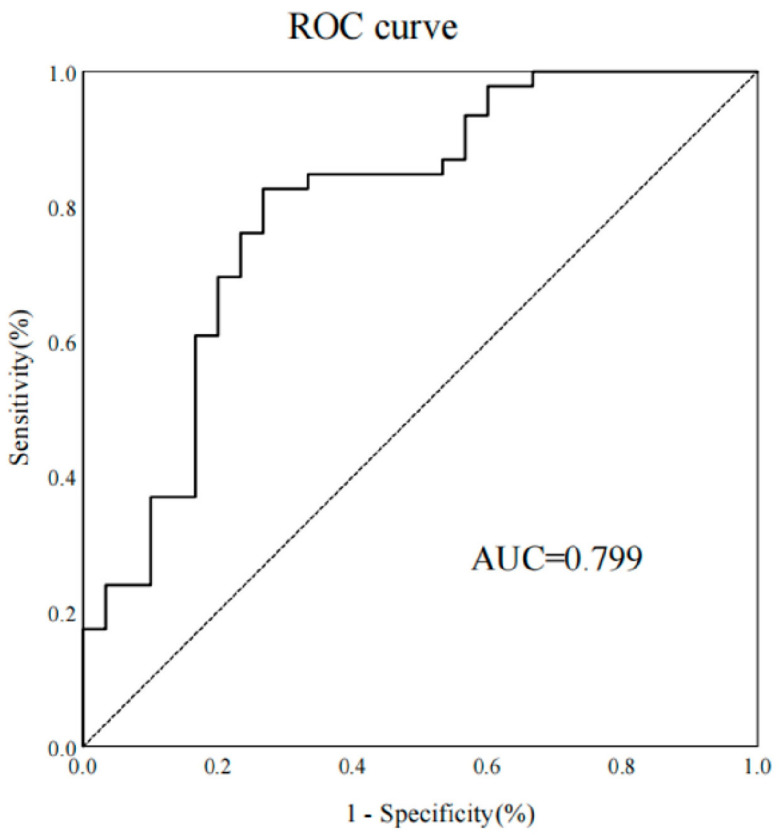
The receiver operating characteristic (ROC) curve was measured using the validation set. AUC, area under the curve.

**Figure 5 jcm-12-00442-f005:**
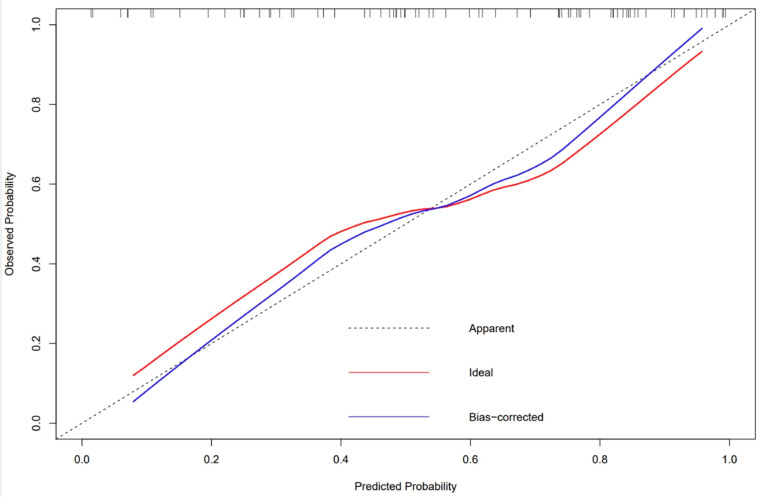
Calibration curve. The red line represents ideal predictions. The plot illustrates the accuracy of the best-fit model (Apparent) and the validation model (Bias-corrected) for predicting vaginal delivery.

**Figure 6 jcm-12-00442-f006:**
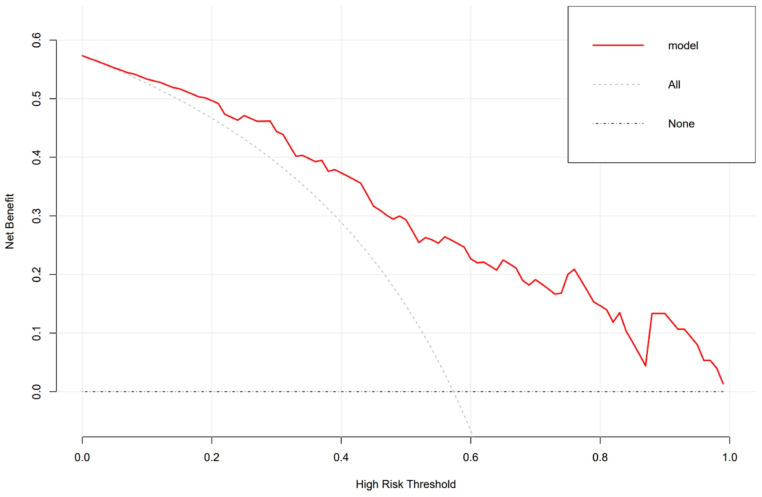
Decision curve analysis for the predictive model. The red line represents the prediction model. The gray dashed line assumes all women delivered vaginally and the horizontal line assumes no woman delivered vaginally.

**Table 1 jcm-12-00442-t001:** The Demographic Characteristics and MRI Parameters.

Characteristic	Training Set (*n* = 75)	Validation Set (*n* = 75)	*p* Value
Maternal age (y), mean (SD)	30.5 ± 3.8	30.1 ± 3.7	0.502
Maternal weight (kg), mean (SD)	66.0 ± 7.1	66.0 ± 7.5	0.999
Maternal height (cm), mean (SD)	157.5 ± 4.9	158.2 ± 5.5	0.378
Gestational age at delivery (weeks), mean (SD)	39.3 ± 0.7	39.1 ± 0.8	0.122
Body mass index (kg/cm^2^), mean (SD)	26.7 ± 2.9	26.4 ± 3.1	0.625
Induction	42 (56%)	38 (51%)	0.624
Cesarean delivery, number (%)	30 (40%)	32 (43%)	0.868
Vaginal birth, number (%)	45 (60%)	43 (57%)	
Fetal biparietal diameter (cm), mean (SD)	9.4 ± 0.3	9.4 ± 0.4	0.603
Fetal femur length (cm), mean (SD)	7.2 ± 0.3	7.2 ± 0.3	0.685
Fetal head circumference (cm), mean (SD)	33.3 ± 1.2	33.2 ± 1.2	0.493
Fetal abdominal circumference (cm), mean (SD)	34.8 ± 1.4	34.5 ± 1.8	0.274
Pubic angle (°), mean (SD)	81.0 ± 6.3	82.6 ± 5.9	0.117
Intertuberous distance (cm), mean (SD)	120.3 ± 10.1	119.3 ± 9.3	0.537
Interspinous diameter (cm), mean (SD)	105.2 ± 7.6	106.0 ± 7.3	0.522
Obstetric conjugate (cm), mean (SD)	123.1 ± 7.9	125.1 ± 9.5	0.169
Bilateral acetabulum distance, mean (SD)	126.2 ± 6.1	127.9 ± 7.2	0.182
Pelvic width (cm), mean (SD)	119.9 ± 6.4	120.6 ± 9.7	0.580
Sagittal outlet diameter (cm), mean (SD)	103.4 ± 6.8	104.6 ± 8.1	0.320
Coccygeal pelvic outlet (cm), mean (SD)	80.7 ± 7.9	79.1 ± 7.8	0.211
Pelvic inclination (°), mean (SD)	51.6 ± 5.0	51.4 ± 4.5	0.825
Sacrum length (cm), mean (SD)	109.7 ± 10.5	112.0 ± 9.9	0.168
Birth weight (g), mean (SD)	3478.5 ± 365.8	3431.1 ± 426.3	0.465

SD, standard deviation.

**Table 2 jcm-12-00442-t002:** The Variables of Vaginal Birth in the Test Set Using Univariate Logistic Regression Analyses (*n* = 75).

Variable	OR	95% CI	*p* Value
Maternal height	1.10	0.99–1.21	0.078
BMI	0.74	0.61–0.91	0.003 ^a^
Fetal head circumference	0.51	0.31–0.84	0.008 ^a^
Fetal abdominal circumference	0.70	0.49–0.99	0.041 ^a^
MRI pelvimetry			
Bilateral acetabulum distance	1.08	0.99–1.17	0.081
Interspinous diameter	1.12	1.03–1.22	0.006 ^a^
Intertuberous diameter	1.06	1.00–1.11	0.045 ^a^
Obstetric conjugate	1.03	0.97–1.01	0.281
Pelvic width	1.03	0.96–1.11	0.463
Sagittal outlet diameter	1.00	0.94–1.07	0.960
Coccygeal pelvic outlet	1.00	0.94–1.06	0.955
Pubic angle	1.08	1.00–1.17	0.060
Pelvic inclination	0.98	0.89–1.07	0.593
Sacrum length	1.01	0.97–1.06	0.599

^a^*p* < 0.05. BMI, body mass index; OR, odds ratio; CI, confidence interval.

**Table 3 jcm-12-00442-t003:** The Predictive Model for Vaginal Birth in the Training Set Using Multivariate Logistic Regression (*n* = 75).

Covariate	Beta Coefficient	OR	95% CI	*p* Value
Intercept	28.755			0.021
BMI	−0.330	0.72	0.57–0.9127	0.007
Fetal head circumference	−0.677	0.51	0.28–0.91	0.023
Fetal abdominal circumference	−0.234	0.79	0.49–1.28	0.342
Interspinous diameter	0.096	1.10	0.97–1.25	0.128
Intertuberous diameter	0.009	1.01	0.93–1.90	0.829

OR, odds ratio; CI, confidence interval. Candidates were recruited by univariate logistic regression analyses and the predictive model was constructed by multivariate logistic regression.

**Table 4 jcm-12-00442-t004:** The Parameters of the Cesarean Section and Vaginal Delivery Groups in prolonged stage of labour (*n* = 54).

Variable	Cesarean Section (*n* = 25)	Vaginal Delivery (*n* = 29)	*p* Value
Maternal age (years)	30.32 ± 3.42	30.00 ± 3.47	0.735
Gestational week	39.36 ± 0.75	39.21 ± 0.62	0.418
Height (mm)	156.80 ± 4.15	158.27 ± 4.90	0.242
BMI (kg/m^2^)	26.82 ± 2.85	26.37 ± 2.74	0.551
Fetal biparietal diameter (cm)	9.63 ± 0.24	9.42 ± 0.32	0.008 ^a^
Head circumstance (cm)	33.83 ± 0.89	33.32 ± 1.11	0.071
Abdomen circumstance (cm)	35.89 ± 1.51	34.57 ± 1.46	0.002 ^a^
Fetal femur length (cm)	7.28 ± 0.16	7.19 ± 0.24	0.143
Pubic angle (°)	80.96 ± 5.77	82.83 ± 5.33	0.222
Intertuberous distance (cm)	119.59 ± 8.41	119.22 ± 8.80	0.875
Interspinous diameter (cm)	104.33 ± 6.37	105.99 ± 5.57	0.310
Obstetric conjugate (cm)	124.58 ± 8.26	124.96 ± 9.38	0.873
CID	18.09 ± 5.70	21.29 ± 5.74	0.046 ^a^
Birth weight (g)	3661.60 ± 324.07	3328.28 ± 329.48	0.000 ^a^

^a^*p* < 0.05. BMI, body mass index; CID, cephalopelvic index of diameter.

## Data Availability

The raw data supporting the conclusions of this article will be made available by the authors, without undue reservation.

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
