# Peer review of "Predictive Value of MRI Pelvimetry in Vaginal Delivery and Its Practicability in Prolonged Labour—A Prospective Cohort Study"

_jcm, 2023, doi:10.3390/jcm12020442_

Round 1

Reviewer 1 Report

  Thank you for requesting  to provide a review of this article, which has a subject of high interest. 

   The main purpose of the analysis was to investigate the probability of vaginal delivery and distinguish the cephalopelvic disproportion risk in women with prolonged active labor, using MRI pelvimetry. The study was a prospective cohort analysis and was conducted for a period of time between 2020 and 2021 in China, with 215 nuliparous women enrolled.

  Regarding the structure and accuracy of the phrases, the manuscript has well structured information, with supported evidence and well structured phrases.

  The manuscript is original and well defined. The results provide an advance in current knowledge. The results are being interpreted appropriately and are significant, as well as the conclusions.

  The article is written in an appropriate way. 

  The study is correctly designed and the analysis is being performed at high standards, so the data are robust enough to draw the conclusion. 

  Surely the paper will attract a wide readership. 

  The English language is appropriate and well understandable.

  There are a few things to add in the lines below, but the article should be published after the corrections are made: 

Line 14: „.” after „process”, not „,”

Line 71: „,” after „labor”

Line 71: which means, not „that means”

Line 72: who should try, not „should try”

Line 72: no matter if spontaneous, not „no matter spontaneous”

Line 87: „,” after „ultrasound”

Line 119: „,” after „variables”

Line 123: „,” after „0.75”

Line 220: „,” before and after „therefore” 

Author Response

Thank you very much and we appreciate you for all the positive comments on our manuscript.

Point 1: There are a few things to add in the lines below, but the article should be published after the corrections are made:

Line 14: „.” after „process”, not „,”

Line 71: „,” after „labor”

Line 71: which means, not „that means”

Line 72: who should try, not „should try”

Line 72: no matter if spontaneous, not „no matter spontaneous”

Line 87: „,” after „ultrasound”

Line 119: „,” after „variables”

Line 123: „,” after „0.75”

Line 220: „,” before and after „therefore”

Response 1: We have made corresponding revisions according to your advice, words in red are the changes I have made in the text.

Reviewer 2 Report

The work is interesting as one step in the search for the ideal way to make an accurate decision on the way to give birth.
Leaving aside the fact that MRI is a very expensive procedure, this test should be considered only as an auxiliary test and not as a decisive one for choosing the way to give birth.
The proposed model does not take into account several complications, such as the bending of the head in the longitudinal head position.
The value of the work would be greatly enhanced by a comparison with traditional pelvimetry methods. Then it would only be possible to state with conviction that MRI represents a significant advance here.
Pg. 1, row 32-33: The sentence is redundant.
The exclusion criteria need to be revised:
Pg. 1, row 73-76: What were the serious complications that made it impossible to give birth by natural means? What were the exponents of intrauterine fetal distress? The exclusion criteria do not agree with Fig. 1 and a priori cesarean section is a matter of course.
Pg. 2, row 87-88: fetal dimensions should be determined on the day of the MRI and not a week before.

Author Response

Thank you for your reviewing and please see the attachment.

Reviewer 3 Report

Please, correct carefully the whole manuscript.

Examples

Line 14: Please, correct “process, We” to “process. We”.

Line 16: Please, correct “labor.Methods” to “labor. Methods”.

Furthermore, with the exception of rare cases of severe pelvic or fetal abnormalities, consider that the grand majority of shoulder dystocia incidents occur in patients with normal pelvic dimensions and it was shown that imaging studies are not useful for identifying patients at increased risk. Thus, related comments should be included in your study. 

Author Response

Thank you very much for your review and we have studied the positive and constructive comments and suggestions carefully.

Point 1: Please, correct carefully the whole manuscript.

Examples

Line 14: Please, correct “process, We” to “process. We”.

Line 16: Please, correct “labor.Methods” to “labor. Methods”.

Response 1: We have made corresponding revisions according to your advice, words in red are the changes we have made in the text.

Point 2: Furthermore, with the exception of rare cases of severe pelvic or fetal abnormalities, consider that the grand majority of shoulder dystocia incidents occur in patients with normal pelvic dimensions and it was shown that imaging studies are not useful for identifying patients at increased risk. Thus, related comments should be included in your study.

Response 2: Thank you for your suggestion, the scenarios like shoulder dystocia incidents were not occur in our cohort, but we agree to your comments and add the shoulder dystocia incidents  in”Strengths and limitations” section.